# A Sustainable Proposal for a Cultural Heritage Declaration in Ecuador: Vernacular Housing of Portoviejo

**Raúl Vinicio Hidalgo Zambrano** [1]**, Celene B. Milanes** [2,3,*]**, Ofelia Pérez Montero** [4]**, Carlos Mestanza-Ramón** [5]**, Lucas Ostaiza Nexar Bolivar** [1]**, David Cobeña Loor** [6]**, Roberto Galo García Flores De Válgaz** [1] **and Benjamin Cuker** [7]

1. Admission and Leveling Institute (IAN in Spanish), Universidad Técnica de Manabí, Portoviejo 130103, Ecuador
2. GeMarc and GESSA Research Groups, Civil and Environmental Department, Universidad de la Costa, Street 58 # 55–66, Barranquilla 080002, Colombia
3. Coiba Scientific Station (Coiba AIP), Gustavo Lara Street, Building 145B, City of Knowledge, Clayton 0801, Panama
4. Multidisciplinary Studies Center of Coastal Zone, Universidad de Oriente, Santiago de Cuba 90900, Cuba
5. Research Group YASUNI-SDC, Faculty of Life Sciences, Escuela Superior Politécnica de Chimborazo, Sede Orellana, El Coca 220001, Ecuador
6. Faculty of Architecture, Universidad San Gregorio, Portoviejo 130103, Ecuador
7. Department of Marine and Environmental Science, Hampton University, Hampton, VA 23668, USA
* Correspondence: cmilanes1@cuc.edu.co

**Abstract:** Vernacular houses treasure the knowledge and traditions of nations. They express the cultural heritage of different generations, including local materials and non-professional designs evolved by resident communities. In South America, vernacular houses often are designed in rural areas. These are influenced by the customs of the indigenous people who inhabited this region for centuries before colonization. In the coastal area of Ecuador, particularly in the canton of Portoviejo, belonging to the province of Manabí, there is an architectural typology called "housing of the three spaces", which has not been valued as cultural heritage. This article responds to the research question of how to structure a sustainable architectural solution, which observes the patrimonial values of the housing of the three Manabí spaces, and which contributes to the resolution of the housing problem in rural Ecuadorian areas. The research was descriptive. The questionary technique was used to characterize these housings and analyze their sustainability criteria and historic heritage values. The results contribute relevant information for the consideration of the housing of the three spaces as cultural heritage. Furthermore, we explored a conceptual and analytical transition of the modern housing named Biosuvernacular (*bio* meaning life, *su* for sustainability and *vernacular* for traditional design) with reasonable economical solutions for resolving the housing problem in the study area.

**Keywords:** housing of the three spaces; vernacular architecture; sustainable architecture; ecological housing; Biosuvernacular

## 1. Introduction

The concept of "heritage" should be understood broader than the usual categorization into object-related "cultural heritage". The concept of heritage includes humanity's tangible and intangible testimonies to be handed down from generation to generation, whether on land or at the bottom of the sea [1]. Heritage is preserved in both objects and living cultures [2]. Cultural heritage not only brings together the collective memories of humankind and the forms of expression of our ancestors, but also positively represents the dignity, uniqueness, and identity of individuals, peoples, groups, and communities living today [1–4].

Various studies examined the conservation of tangible and intangible cultural heritage of nations and peoples. These include solutions for the conservation of cultural heritage [4],

digital management methods [5,6], the need for funding heritage projects [7], indicators of vulnerability and risk [8], as well as multi-threat studies of cultural heritage [9–11]. For the present work, progress is also being made in creating theoretical frameworks for conserving vernacular houses based on values [12,13].

The Agenda for Sustainable Development (SDG 2030) [14] calls for establishing objectives that mobilize nations' social and economic transformations that innovate new ways to achieve Sustainability. Among these, SDG number #11 aims to make cities and human settlements inclusive, safe, resilient, and sustainable in Latin America and the Caribbean. For its part, SDG #9 proposes to build resilient infrastructures, promote inclusive and sustainable industrialization, and encourage innovation in Latin America and the Caribbean. These are the main objectives to which this research contributes, with its focus on vernacular housing.

In the broadest sense, vernacular housing refers to dwellings built of local materials to designs evolved by many generations of residents. These are built in the absence of formally schooled architects and engineers. Such buildings dominate the world's stock of housing, particularly in the developing world. Different authors have conceptualized and reinterpreted the concept of vernacular housing. Some reveal regional sociocultural factors' role as decisive forces shaping the traditional living space and its built manifestations. [15,16]. Recent studies suggest the existence of "five trends that consider vernacular architecture as an aesthetic object, a biological phenomenon (types and evolution), a material-physical substance (physical explanations), a cultural-sociological reality, and, finally, an anthropological phenomenon" [17,18]. Other authors have contributed to the evaluation of typologies of these housings [19].

The review of the scientific literature allowed us to argue as a hypothesis that the proposal of guidelines, from sustainable architectural solutions that respect the patrimonial values of the vernacular housing, will contribute to the rescue and conservation of the housing located in rural areas from Portoviejo.

However, the vernacular dwellings in the rural area of Portoviejo, Ecuador, lack appreciation as cultural heritage, as well as their recognition and management by the competent authorities in the country. They also lack proposals for technical solutions that resolve the complex housing situations for their population.

In this article, a proposal for a declaration of cultural heritage arises, defining its concept as a certain number of tangibles. These intangible and natural goods constitute part of social practices, to which values are assigned to be transmitted from one generation to another one. Therefore, for a property to be declared a cultural heritage in Ecuador, in addition to its Outstanding Universal Value, it must be unique and have authenticity characteristics and adequate management for its conservation and protection.

This research focuses on the study of *vernacular housing of three spaces*. These spaces are (kitchen, bedroom room, and circulation corridor). The analysis is carried out from the perspective of cultural heritage and sustainable development, setting guidelines for recovery, and social recognition of these singular houses. It responds to the scientific question of how to structure a sustainable architectural solution, which observes the heritage values of the housing of the three spaces of Manabí. This research contributes to solving the housing problem in rural Ecuadorian areas.

## 2. Materials and Methods

The research was descriptive with a qualitative approach. In its first phase, the content analysis technique was applied in relevant journals of high national and international impact on the history and evolution of the "houses of the three spaces" in Ecuador (Figure 1). The theoretical framework of the research was designed to integrate the approaches of cultural heritage and sustainable development.

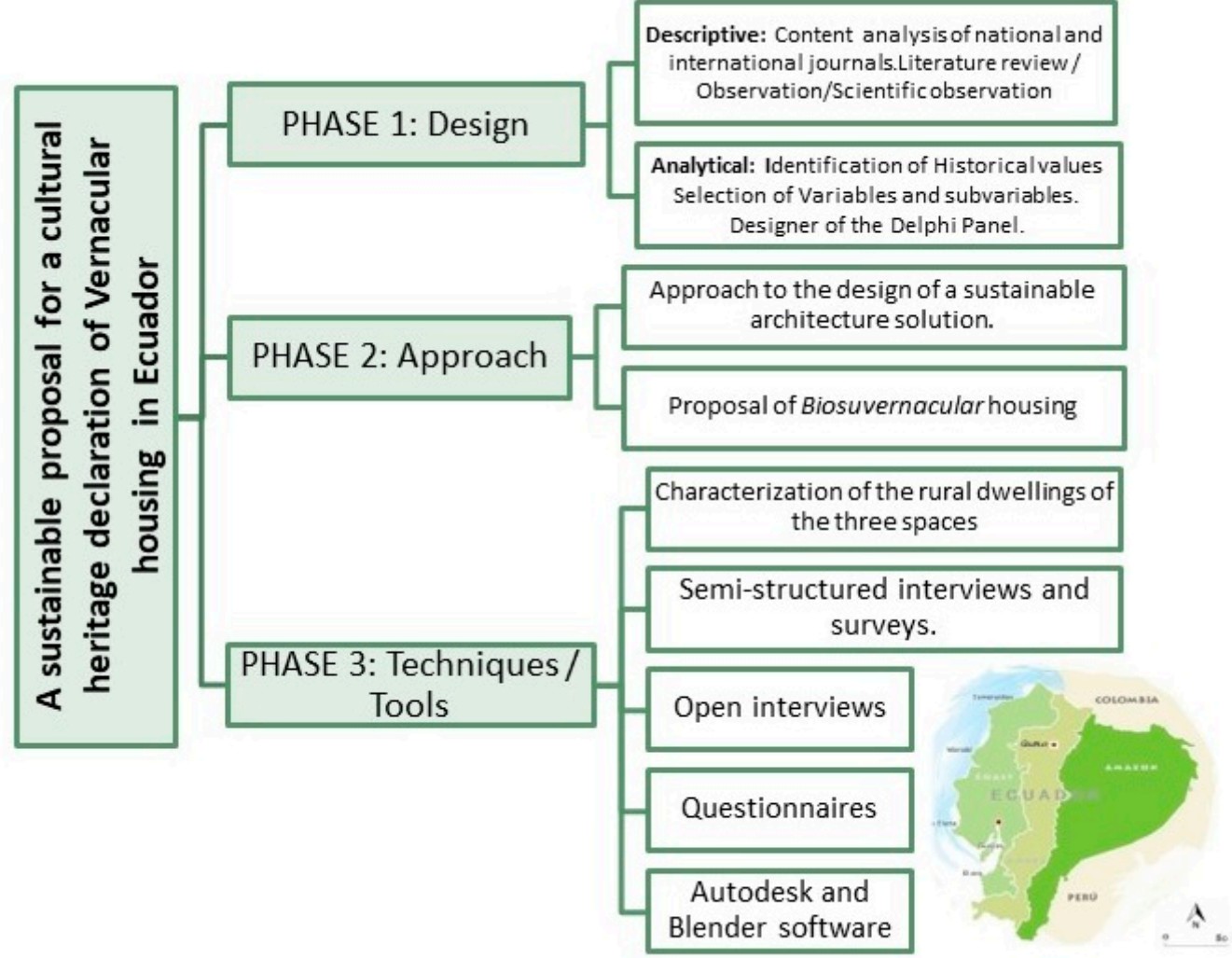

**Figure 1.** Methodological graph of the research.

Phase 1: Scopus and the Web of Science databases were consulted to identify the diversity of terms and concepts regarding vernacular architecture. The defined search parameters comprise published literature on the subject, emphasizing the last five years. At the same time an expert driven Delphi Panel [20] was designed and implemented as a methodological instrument to validate the proposal and design of an architectural solution that observed the heritage values of housing in the three accessible spaces for rural communities in Ecuador. The suggestion of a new analytical term for a *Biosuvernacular* house was submitted to expert criteria, as well as the proposal of the dimensions, categories, variables, and indicators that allowed arguing the heritage values of the vernacular house and its implementation in the new sustainable architectural–tectonic solution. We developed an evaluative scale using the following criteria, High Incidence (3), Medium Incidence (2), Low Incidence (1), Null Incidence (0), and No Criteria (sc), to evaluate the correspondence of the concepts, categories of the proposed architectural solution with the cultural heritage, and sustainable development approach. The relevance quotient was applied for the selection of experts. Twenty-one experts were selected and classified as having a high coefficient of expertise in the reference topic.

Those variables and sub-variables that obtained between 80% and 100% approval and high incidence evaluation by the experts were considered valid. The variables and sub-variables considered were: (a) economic value (accessibility, technology, pre-budget, and construction techniques); (b) aesthetic value (composition, harmony, texture, color, location, and originality); (c) historical value (three spaces, historical evolution, traditions,

and construction system); (d) use value (satisfaction of needs, tangible value, intangible value, and habitability); (e) formal value (bioclimatic elements, structural aspects, visual attraction, and architectural aspects); (f) symbolic value (design elements and links between past and present).

Phase 2: This involved the characterization and diagnosis of the rural dwellings in the three Manabí areas. The techniques of scientific observation [20], interviews, and semi-structured surveys [21] were applied. These allowed us to obtain information about the vernacular houses in the communities of the seven rural parishes of the Portoviejo canton. To diagnose the state of the vernacular dwellings in the rural communities of Manabí and to identify the cultural value attributed to them by their inhabitants, the survey and scientific observation considered the following variables: (a) socio-demographic data of the family living in the dwelling (number of members, age, sex, occupation); (b) data on the dwelling (year of construction, structure, types of materials, walls, roofs, floors, function of interior spaces, and type of maintenance); (c) data on the environment (characteristics of the surroundings, use of exterior spaces, use of natural resources of the environment); (d) resistance of the dwelling (exposure to anthropic or natural risks).

An interview was conducted to obtain relevant data on the life histories of the families related to the following variables: (a) construction history of the dwelling (how many generations have lived in it, who and how they decided to build it); (b) the origin of the dwelling (built by the current family or inherited); (c) socio-anthropological aspects (on the use of spaces, distribution of functions, relationships with other relatives and family members in the construction, relationship with the natural resources of the environment); (d) cultural aspects (perception of the population on the values of the dwelling, the meaning of its spaces, the relationships of its members with the spaces).

Phase 3: A sustainable architectural solution was designed. The experts approved the following variables and sub-variables as having a high impact on the proposed architectural solution between 70% and 100%. These are (a) vernacular design (functional: three spaces, stilt construction, progressive design); (b) structural design (light structure, seismic-resistant, efficient against natural and anthropic phenomena); (c) bio-vernacular proposal (passive bioclimate, clean energy, endemic vegetation); and (d) budget (accessibility, unit price analysis, local labor, renewable materials, and environment). Professional software including AutoCAD 2D and 3D, Autodesk, and Blender were used to model the architectural proposals of the study.

The sample. A total of 309 houses were selected for the study from the seven rural parishes of the canton of Portoviejo. This represents 100% of the total number of homes in the three areas of Portoviejo. A total of 309 people were surveyed, one for each housing unit. Of these, 80% were women and 20% were men. Occupations of residents included 55% farmers, 15% rural workers, 5% merchants, 15% unemployed, and 10% occasional workers.

*Study Area*

Province of Manabí in Ecuador is localized in the center of the nations' coast. With an area of 18,831 km$^2$ the province has had great historical importance due to its conditions and resources. The city of Portoviejo, in the canton of the same name, is the capital of Manabí (Figure 2). It is a territory where primary economic activities such as agriculture, livestock, and fishing are developed. The fishing industry is centered in the Manta canton, emphasizing tuna. The extensive and beautiful beaches support a tertiary sector based on commerce and tourism [22].

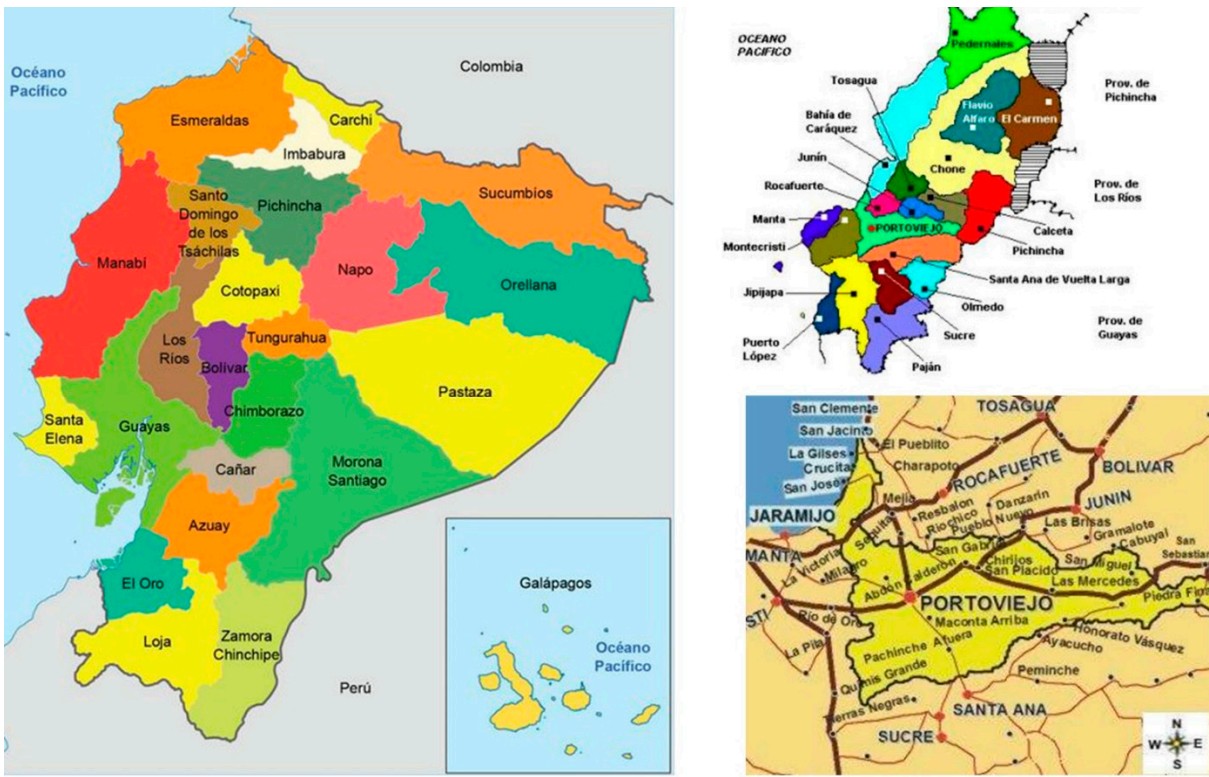

**Figure 2.** Location of Ecuador, Manabí, and Portoviejo as capital of Manabí province.

In the 2010 census, the canton of Portoviejo had a population of 206,682 inhabitants. It is the eighth most populous city in the country, making it one of the main urban centers of the nation.

Four hundred thousand eight hundred seventy-nine housing units were registered in Manabí. A typology was used to differentiate housing solely based on construction materials and urban residence. What we know as a "vernacular house" is generally cataloged as "rancho." That is, "a rustic construction, covered with palm, straw, or any other similar material, with walls of cane or bahareque (adobe like walls of mud reinforced with cane or sticks) and with a floor of cane, wood or earth." (Figure 3) In the case of Portoviejo, vernacular houses' construction characteristics, as in other regions of Ecuador, reflect: location, aspirations of the resident, a spectrum of variants, architectural conception, building methods, and their insertion or transformation of the context [23].

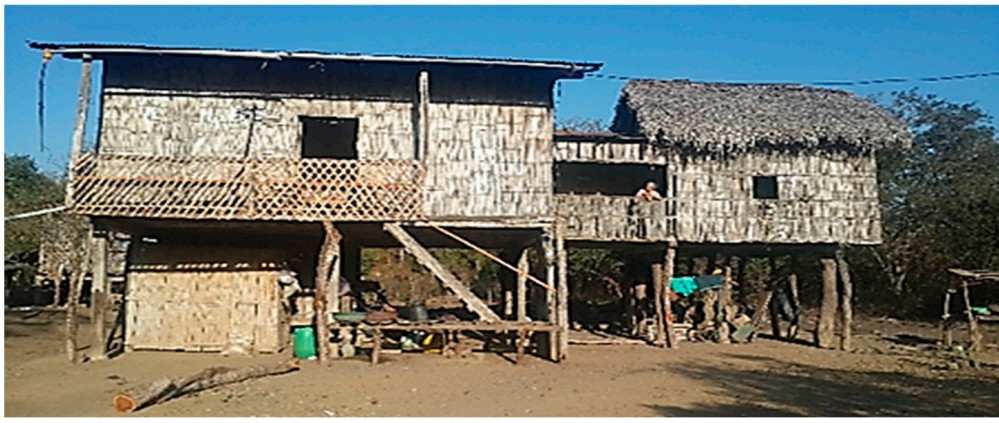

**Figure 3.** Manabí house of the three spaces. Located in Calderón Parish. Portoviejo. Source: Photo taken by the author.

## 3. Results

### 3.1. The Concept of Vernacular Housing

Throughout history, several authors have defined the term "vernacular architecture." Although there is no agreed definition for vernacular architecture, the one proposed by the International Council on Monuments and Sites [3], in a general sense, speaks of "a continuous process, which includes necessary changes and constant adaptation in response to social and environmental requirements". It is a form of architecture strongly marked by an environment [24], establishing a point of view in the use of design skills and the tradition of local builders. More recently, vernacular architecture has been examined by designers and the building industry to become more energy conscious with contemporary design and construction as part of a broader interest in sustainable design [25].

A summary analysis of the published literature generally characterizes vernacular architecture in different ways [23,25–27]. The theoretical approach to the concept of vernacular architecture is broad, and it has been collected in texts such as Bernard Rudoksky 1964 [1,28]. The "Charter of Built Vernacular Heritage" by ICOMOS [3] considers Vernacular Architecture as "the fundamental expression of a community's identity, in its relations with the territory and at the same time, the expression of the world's cultural diversity".

Other concepts have also been developed in response to political realities or programs in different countries to address the issue of traditional housing. The following terms stand out: Vernacular Urban Housing [29], Housing of Social Interest [30], Housing of Priority Sustainable Interest [31], Sustainable Housing [32,33], Traditional Housing [34], Contemporary Rural Housing [34], Social Interest Housing [35,36], Subsidized Housing [37], Vernacular Construction [38], Sustainable Modular Housing [38], and Sustainable Social Interest Housing [39]. The authors made an extensive review of concepts that are reflected in Appendix A.

The perspective of this article reflects the approaches of the UNESCO Charter on Vernacular Housing, aligned with the Ecuadorian norms on the nation's cultural heritage [40]. This is the starting theoretical base to characterize the values of Manabí vernacular housing present on the Ecuadorian coast since the eighteenth century.

The Historical Evolution of the Spaces in the Vernacular Housing of the Three Areas of Portoviejo

The historical investigations on the evolution of housing in Ecuador reveal that during the beginning of the colonial period, there were transformations in the design and construction of vernacular housing in the area that today occupies the province of Manabí. In this period, the influence of the Spanish conquerors became clear: rural vernacular dwellings with stilts and upper structures with internal organization appeared, similar to the wooden houses of Europe [23]. Vernacular housing constructions of recognized value have been developed from the pre-Columbian, Colonial, and Republican periods to the most recent times. Builders passed the construction tradition down from generation to generation. They evolved over time, integrating practice, and good workmanship. The previous invites us to address the values of vernacular housing in Manabí from the different historical stages through which the evolution of its design and construction passes.

The pre-Columbian autochthonous (of local origin) vernacular architecture comprises different periods according to the stages of Ecuadorian prehistory [23]. It started in the Preceramic (9000–3500 BC) with circular single-family homes characterized by their beehive shape with a rustic wooden structure anchored in the ground, covered with bamboo and straw. The Formative period (3500–500 BC) presented community dwellings with an oval geometric shape, made of wooden posts, with one of these as a support for the roof in the center of it, and sizes that could reach $8 \times 12$ m.

Corresponding to the Regional Development period (500 BC–500 AD) [41], bahareque houses were made of sticks interwoven with vines and covered with clay. At the same time, the houses of the Integration period (500–1500 AD) made use of the guadua (thorny clumping bamboo) cane lattice and the presence of stairs that would indicate the existence

of an upper floor in the houses. After the arrival of the European conquerors, the places acquired two typologies: the quadrangular single-family home on a basement and the quadrangular single-family house "on stilts". In the colonial period, vernacular rural dwellings essentially maintained their traditions, using ecologically inexpensive materials for the indigenous people. The Spanish introduced single-family houses with two floors, outlining a typology of urban housing of the colonizer or his direct descendants, which was based on Arab-Spanish architecture, yielding the use of galleries and the centralization of the inner courtyard.

These constructions had adequate lighting for the house, primarily through slots in the walls and high spaces, which also facilitated circulation of air. This allowed for fresh breezes in the hot and humid climate prevailing in the Manabí region [42]. In addition to its ecological nature, it was built with cane, bamboo, wood, and straw structures that facilitated numerous openings that functioned as air-breathing elements. Such housing persists even in our times in rural areas. This style of construction and architecture became a cultural tradition, to the point of being maintained and built up to the present day. The new environmental and natural risks that affect the study area have raised the need to seek construction solutions to address longer periods of rain and flooding, both river and coastal, common in the low-lying regions present in the rural area of Portoviejo.

In this context, the Spanish founded and developed consolidated cities with a type primarily known as urban centrality. On the outskirts were the homes of the native population, who maintained, in their houses, the characteristics reflecting their traditions [23]. The dominant institutions gathered around the central square were the Church and the Cabildo, or the Town Hall, whose officials had the mission of consolidating Spanish rule. In this period, the lots were distributed in the following order of hierarchy: the Spaniards, their descendants, and the natives. Such settlements functioned to further colonize and guarantee the domination of the indigenous people by the Spanish.

The three-space dwelling consists of an environment for rest, an environment for the kitchen, and a space (bridge) that joins these two. This bridge is intended to isolate, in the event of a fire, the kitchen from the rest of the house [43]. Over time this bridge evolved into a dining room. On the ground floor were the areas to store grains, the warehouse, and the rest area.

The evolution of the home environments of the three spaces is marked by the needs that appear over time. The influence of "new materials" used in construction by the Spanish colonizers (lime, clay, sand, and iron nails) [23] also reached the rural area of Manabí. There, vernacular houses with three spaces continued to be built with wood, bamboo, and other natural materials, and places with masonry walls were introduced by families with greater wealth. In this area, the custom of building homes on stilts prevails, in many cases, using the lower part for commercial activities.

Today, few houses entirely conserve the use of renewable materials from the environment. The evolution of the dwelling in the three spaces occurred both in the use of materials for its construction and in the use of the rooms or openings of the home. The ground floor, which was left undivided to be used as a rest area and protection against animals and floods, was closed to be used as a warehouse, corrals, trading area, or a sales area for homemade products. The resting room, dining room (before the bridge or corridor), and the kitchen are on the upper floor. The toilet service area has always been considered outside the home. More modern construction moved the toilet to a portion of the resting room or to the rooftop [43].

The dimensions of the rest environment have also increased. Before, there was only one bedroom. Currently, the number of bedrooms reflects the size of the house. The corridor was converted into a social area (dining room or living room), and a roof terrace was adapted to the kitchen environment that can serve as a laundry room and space to hang clothes. This type of housing is still being built as part of the Manabí identity and tradition, conserving the three areas but changing the construction materials. Now, indigenous materials are combined with other materials such as brick, block, reinforced concrete, zinc,

asbestos, aluminum, glass, metal sheets, etc. These new materials increase the home's energy consumption during construction and subsequent maintenance.

　Currently the Portoviejo canton has seven rural parishes, which are: Crucita, Río Chico, Pueblo Nuevo, Calderón, Alhajuela, Chirijos and San Plácido. In six of them, 309 dwellings were found to preserve the three spaces' shape [43,44]. Today, there are new construction elements such as, arcades in the front part of the house that stop the sun in the summer but let it enter in the winter, grass roofs, cross ventilation, and other techniques that bioclimatic architects have rescued from collective memory [23]. It was industrialization, the mass construction of housing in the urban environment, and the abundance of fossil resources that led to the abandonment of these centuries-old practices. However, in the context of climate change and the technological advances of the 21st century, there is a need to return to sustainable architectural solutions based on renewable energies and using materials from a bioclimatic approach [23].

### 3.2. Characterization of the Vernacular Housing of the Three Spaces in the Rural Parishes of the Portoviejo Canton

　The questionnaire application revealed the characteristics of the vernacular housing currently in the studied parishes. The parish of Calderón is the most representative of this type of housing. The features of the dwellings of the three spaces in the studied parishes are summarized in Table 1.

　According to the data obtained from the survey, it was possible to contrast the followings relevant aspects of these housings:

　The vernacular dwellings with the highest rate of antiquity are between 11 years old to 40 years old (67%). Dwellings of 41–50 years (14%) and those 51–110 years old (19%) account for the rest. The previous supports the antiquity and sustainability of this type of housing with materials that vary from bamboo, cane, and wood, in addition to zinc sheeting as a cover and, in some cases, the addition of concrete.

　The risks identified by the population surveyed in the study area were river floods (38%), landslides (13%), and rockfalls (4%). That 45% of respondents did not identify any type of risk for their home shows the effectiveness and environmental adaptability to the environment of this architecture.

　The housing of "the three Manabí spaces" include passive climatic principles such as orientation, windows, doors, and other openings to take advantage of natural light, shade, and ventilation. This is accomplished with the use of renewable construction materials, the use of natural elements for thermal insulation, and the application of ancient techniques for its construction.

　Its design responds to climate changes and human protection needs without excessively altering the environment. Trees protect it and, as it is built on stilts, it does not interrupt the paths of animal migration or light river circulation during the winter. It uses the principles of passive bioclimatic control since the structure alone performs the tasks of climatic comfort, use of natural light, winds, etc. This approach eliminates the need for energy consumption to regulate the interior climate and the attendant production of greenhouse gasses and other pollutants.

　Different types of materials were found in the construction of vernacular dwellings. Among them, a predominance of reinforced concrete was wood (35.6%, 17.8%), masonry (16.12%) of homes, and others (1.3%) (Figure 4).

　The type of uses that the population gives to the lower spaces of the Manabí vernacular house was also identified. The most recurrent were living areas (19.8%), rest areas (35.3%), cellars (16.5%), housing (14.9%), without use (6.8%), shops (3.9%), and garages (2.9%). Each of them is illustrated below in Figure 5. Types of uses of the lower space in the dwellings of the three spaces include: (a) use as a living area, (b) rest area, (c) trade area, (d) warehouse area, (e) housing area, (f) unused, and (g) garage area.

**Table 1.** Characteristics of housing of the three spaces for parishes studied.

| Characteristics | Most Significant Image of the House |
| --- | --- |
| Parish: Abdon Calderon<br>Extension: 136 Km$^2$<br>Communities studied: 20<br>Number of houses: 108 Population: 14,164 | 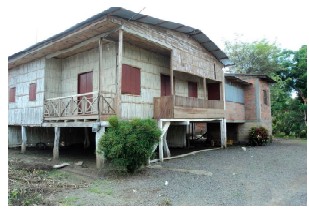 |
| Parish: New Town<br>Extension: 51.40 Km$^2$<br>Communities studied: 13<br>Number of houses: 30 Population: 3169 | 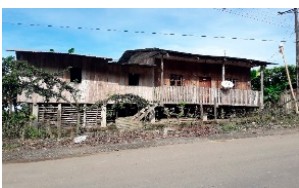 |
| Parish: Rio Chico<br>Extension: 82.81 Km$^2$<br>Communities studied: 17<br>Number of dwellings: 62 Population: 11,757 | 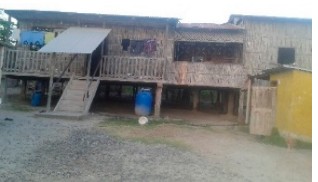 |
| Parish: San Placido,<br>Extension: 216.61 Km$^2$<br>Communities studied: 39 Number of houses:<br>51 Population: 8351 | 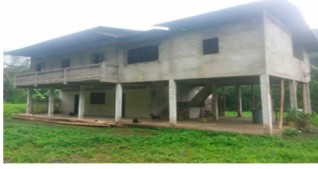 |
| Parish: Chirijos,<br>Extension: 16, 13 Km$^2$<br>Communities studied: 13<br>Number of houses: 13 Population: 2362 | 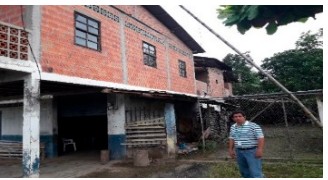 |
| Parish: Alajuela.<br>Extension: 23.20 Km$^2$<br>Communities studied: 10<br>Number of houses: 45 Population: 3754 | 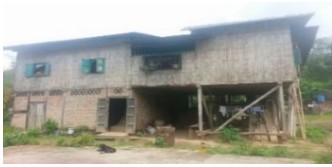 |

The structure of most of the houses in "the three spaces" is made of wood, bamboo, bahareque, equine, or zinc on the roofs. Therefore, it is a light structure that allows it to withstand the discharge of an earthquake or telluric movement as it does not have a greater mass in its consistency. It is also a stilt house. That is, it is not located at ground level because the area where it is built is subject to flooding. This prevents the collapse of the structure in the event of floods and landslides.

The characteristics described above allow us to affirm that the housing of "the three spaces" of Portoviejo has elements of sustainability. The preceding provides managing, based on contemporary architectural trends and solutions based on these elements.

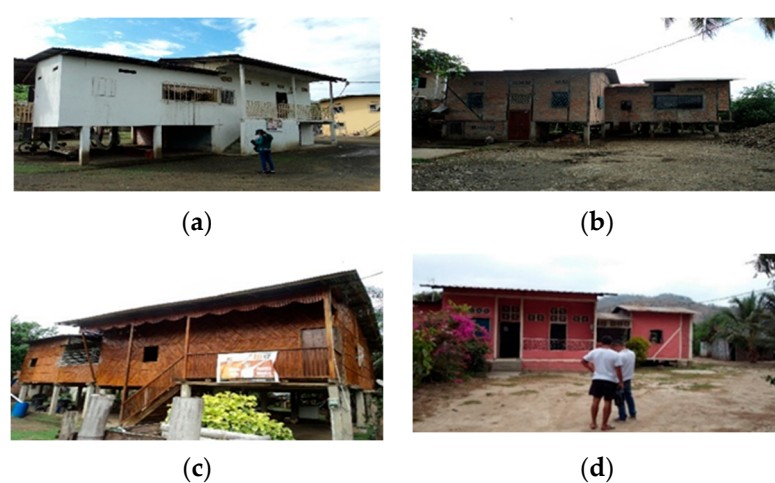

**Figure 4.** The construction materials used in the housing of the three spaces. (**a**) Reinforced concrete, (**b**) masonry, (**c**) wood, (**d**) others.

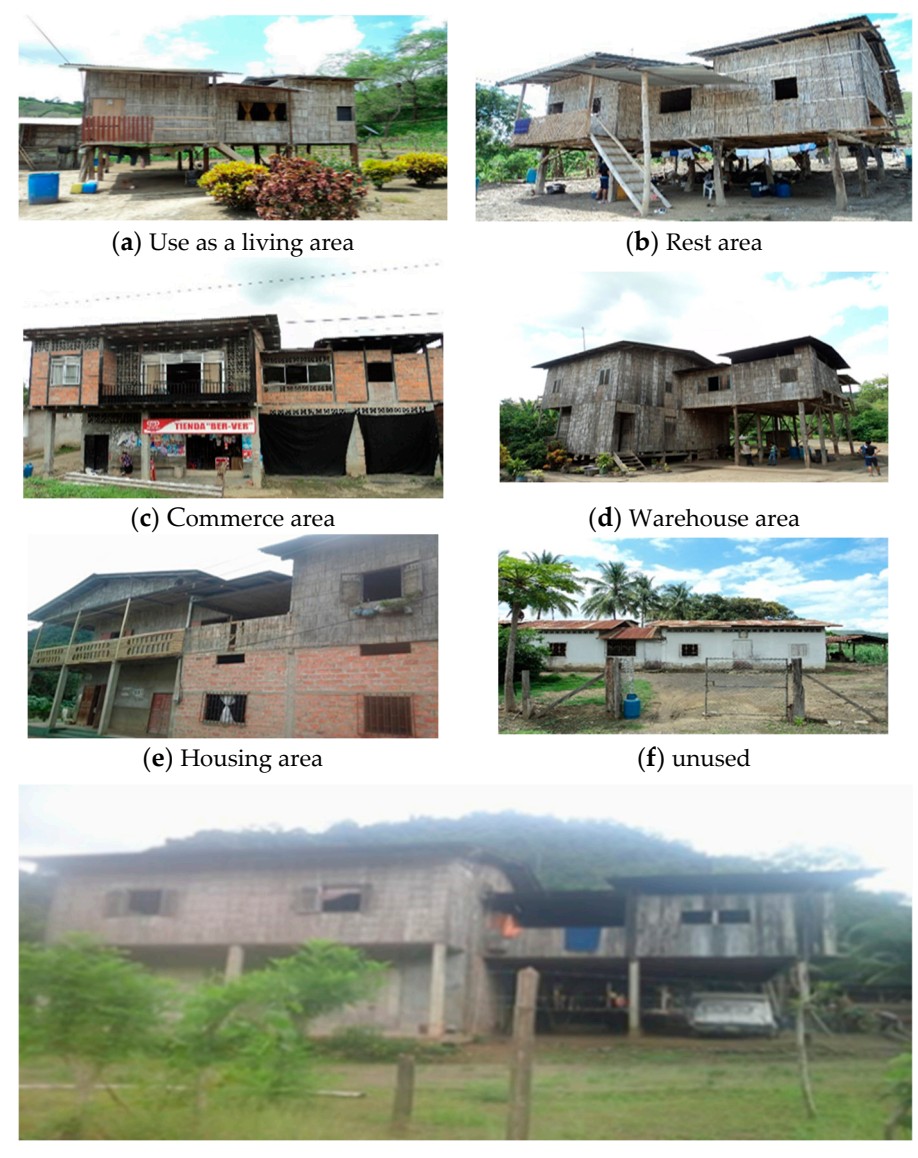

**Figure 5.** Types of uses of the lower space in the dwellings of the three spaces (source: authors).

### 3.3. Proposal for the New Housing Design of the Three Spaces with Sustainability Criteria—The Biosuvernacular House

Our research proposed an analytical transition from the concept of vernacular housing to Biosuvernacular housing, with the aim of offering sustainable technical and architectural solutions that enhance the housing of the three spaces, thus favoring the process of recognition of the population and institutions as cultural heritage. This could be the first step for its management and conservation by communities and cultural institutions.

Understanding the concept of Biosuvernacular housing as one that preserves the historical, cultural, symbolic, aesthetic, economic, and use values of vernacular housing, uses in its construction the principles of passive bioclimatic and environmental sustainability, and includes the use of modern technology and innovation along with the ancestral valuing of the homes of the three Manabí spaces. Bio is taken from the Greek word bios, which means life; su denotes the sustainability of housing; and vernacular comes from the Manabí vernacular housing, which has existed since the 18th century.

In the Biosuvernacular housing proposal, four variables are considered based on indicators defining the element's characteristics:

(a) The first variable is the vernacular design of the new proposal. It is based on a three-room dwelling dating from 1736 [45]. In this design, the three spaces are defined: kitchen and rest linked to the disposable corridor in case of fire in the kitchen area. It will be on stilts, as a response to natural and anthropic phenomena and of progressive characteristic, that is, it will be expanded according to the needs of family space.

(b) The following variable is the structural design of the dwellings. They are characterized by being a light structure, earthquake-resistant, and, therefore, efficient in the face of natural or manufactured threats, including earthquakes, floods, landslides, and fires.

(c) As a third variable and the primary indicator, it is the use of passive bioclimatic control, characterized by the use of the elements of the house and external ones to improve the thermal comfort of the environment in the upper and lower part of the house. They will combine this with the use of clean energy to optimize the use of energy-consuming resources. In addition, they will use the endemic vegetation of the area as an element to maintain a pleasant temperature in the environment and inside the house.

(d) The last variable is the budget, the turning point in any project. This proposal considers the use of renewable construction materials and the environment. As well as the use of qualified local labor, the analysis of unit prices will allow the proposal of a cost accessible from an economic point of view, mainly for the inhabitants of the rural area.

Based on the previous and according to the requirements of Ecuadorian regulations, we propose an architectural variant of a three-space dwelling illustrated below. (Figures 6–8). Figure 9 compares the variables and indicators in the housing of the three spaces and the Biosuvernacular dwelling.

### 3.4. Proposal of Guidelines for the Conservation and Rescue of the Biosuvernacular House of the Three Spaces as Cultural Heritage

We know the challenge of mobilizing the actors to conserve and rescue housing in the three spaces when they are not recognized or managed as cultural heritage. That is why the Biosuvernacular housing proposal articulates guidelines that allow guiding actions. Within the context of this research, the guidelines express the trend for the conservation and rescue of the Biosuvernacular house of the three spaces as cultural heritage. Each guideline contains an objective. These four guidelines are orientations. This proposal was discussed with the Ministry of Heritage, Academy, and College of Architects in Ecuador, among other stakeholders (Figure 10).

The first guideline promotes integrating the results of science into decision making. The actors of the academies and research groups of different levels and scopes in the region can provide results in vernacular housing, its characterization, and diagnosis in the study

area. At the same time, it aims at the second guideline at rescuing heritage values to preserve traditions in current housing proposals.

The third guideline summons all public and private actors in initiatives for the conservation and rescue of heritage. Critical actors in the study area are the Ministry of Culture and its institutions at different levels, local governments, and communities in the rural area of Portoviejo, Higher Education Institutions (IES), and the National Institute of Cultural Heritage (CPI).

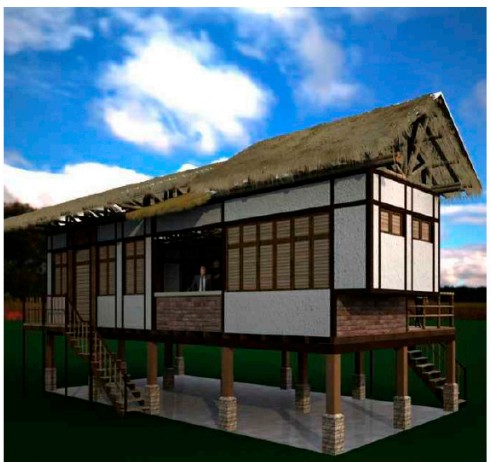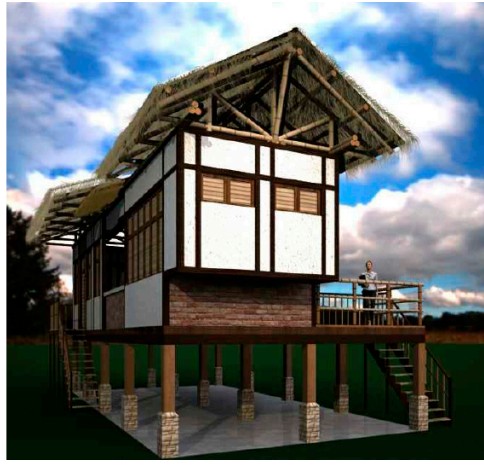

**Figure 6.** Views in spatial perspectives of the three spaces' architectural housing variants were achieved with professional software such as, Autodesk and AutoCAD (source: by the authors).

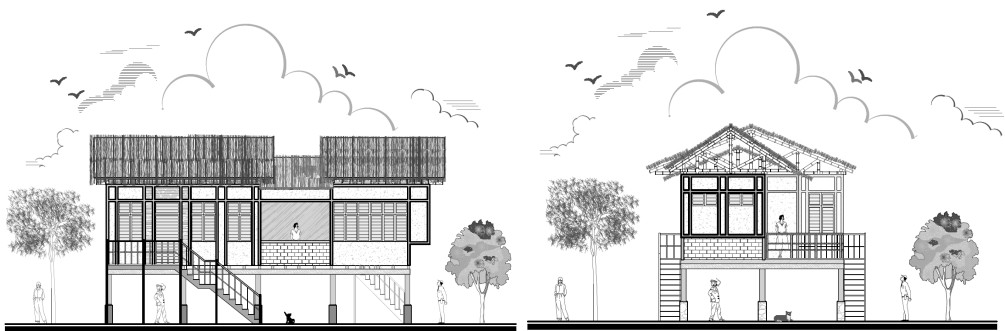

**Figure 7.** Views of the front and side façade of the house with its mobility and setting details. Source: prepared by the author.

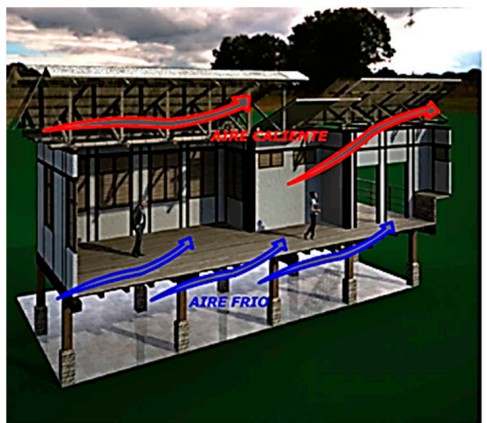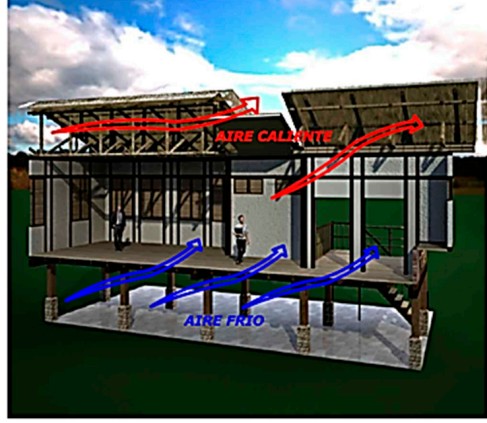

**Figure 8.** Section views with wind circulation. Source: prepared by the author.

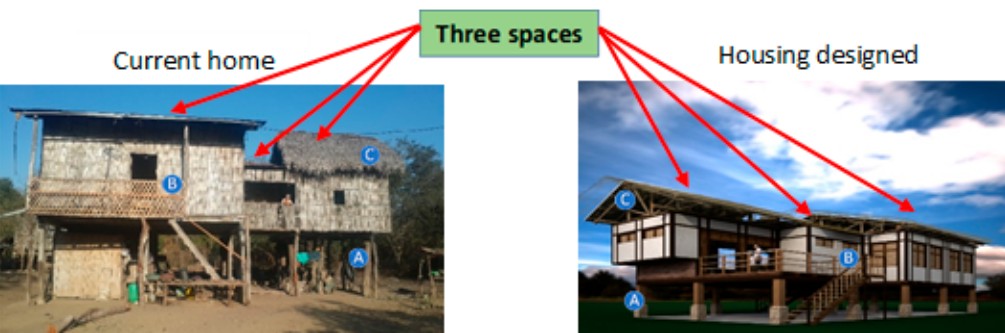

**Figure 9.** Graphical comparison between the housing of three spaces and Biosuvernacular housing. Caption: (**A**) On stilts and on the ground floor; (**B**) use of wood, bamboo, toquilla straw, and bahareque in part of the structure and roof; (**C**) pyramidal light cover.

## Guideline 1

Integrate the results of scientific research in the housing management of the three spaces as cultural heritage.

Objective: To provide scientific information based on diagnoses, monitoring and others to the asset management of housing in the three spaces.

## Guideline 2

Respect the customary law of rural populations in the use, maintenance and construction of the houses of the three spaces.

Objective: Rescue and preserve heritage values (historical, symbolic, use. Formal, and economic)

## Guideline 3

Integrate public and private actors in Portoviejo's cultural heritage management processes, observing the legal framework established by the INPC.

Objective: Empower the public and private actors of the rural communities of Portoviejo for their active participation in the heritage rescue and conservation process.

## Guideline 4

Propose sustainable architectural solutions that respect the nature-housing relationship of the three spaces, safeguarding the cultural heritage built by different generations of rural communities in Portoviejo.

Objective: To propose sustainable and innovative architectural solutions based on bioclimatic approaches to risk management in the face of natural and anthropic threats, for the rural area of Portoviejo.

**Figure 10.** Biosuvernacular housing conservation and rescue guidelines.

The fourth guideline directs architectural proposals and solutions based on passive bioclimatic, sustainability, and economic access to these for rural populations. It is the transition from the vernacular housing of the three spaces to the Biosuvernacular housing with ecological and bioclimatic criteria.

These guidelines, in turn, are based on the criteria of the Ministry of Heritage [40], which indicates that "the value of heritage is to be the channel to relate people to their past and knowing this past, understand the development and the behavior of the present". However, we consider the material part more critical; the intangible heritage is immersed in the house of the three spaces since it has its ancestral construction techniques and the meaning of each home space for its members.

The new proposal validated the following values in its design:

- **Economic value**. The proposal observes the financial accessibility of the rural population to this type of housing. (Validated by experts between 70% to 90% of high and medium incidence in the proposal). The components of a construction budget were analyzed, such as the type of materials to be used, the transportation of materials, the use of skilled or empirical labor, (unskilled made by the experience), the use of minor or significant tools, and direct or indirect costs.

In the direct costs, the price of the material to be used, the use or not of a minor or significant tool, the transportation of the material, and the cost of labor were considered. In the indirect costs, the percentage of the constructor's utility was considered, which in this case is omitted because it is a non-profit project. The rate of contingencies, which is 5%, will not be considered either What should be regarded is the Value Added Tax (VAT) which is 12%. The acquisition of Guarantees for reasonable use of the advance payment and completion of work is also mentioned but not considered.

The proposed budget is USD31,480.16. The proposal for the patrimonial dwelling has USD199.88 m$^2$, which gives it a value of USD157.49 per m$^2$, a cost that, when compared with that of a house constructed from concrete, which fluctuates between USD250 and USD400 m$^2$, presents a difference of 40% below the minimum price. Building with reinforced concrete means the creation of greenhouse gases, environmental pollution, and the consumption of large amounts of energy (see Table 2).

**Table 2.** Comparison of the Biosuvernacular housing budget and the reinforced concrete housing. Source: prepared by the author.

| Housing Type | Total m$^2$ | Value per Square Meter of Construction | Final Budget. |
|---|---|---|---|
| Biosuvernacular. | 199.88 m$^2$ | USD157.49 | USD31,480.16 |
| Reinforced Concrete. Level 1 | 199.88 m$^2$ | USD250.00 | USD49.970 |
| Reinforced Concrete armed. Level 2 | 199.88 m$^2$ | USD400.00 | USD79.952 |

- **Aesthetic value**. The proposal balances composition, harmony, texture, color, construction materials, and originality, preserving the identity and the same materials with which they were built since the 18th century. (Validated by experts between 70% to 90% of high and medium incidence in the proposal).
- **Historical Value**. The proposal preserves and bears witness to family and community history events, maintaining the spaces that tell the most significant events of its inhabitants throughout their life stories since 1783. (Validated by experts between 90% and 100% high and medium incidence in the proposal).
- **Use value**. The proposal maintains the use value of its three spaces and the lower space of the house. It has intangible values associated with its inhabitants' daily life, culture, and customs. As well as tangible matters related to the uses of subsistence and family care. (Validated by experts between 80% to 100% of high and medium incidence in the proposal).
- **Formal Value**. The form of the new proposal observes the bioclimatic elements, structural aspects, visual attraction, and architectural aspects and is pleasant to view. (Validated by experts between 80% to 100% of high and medium incidence in the proposal).
- **Symbolic Value**. The elements of the design are adjusted to be a link between past and present. It has a direct relationship with its builder or designer and with its use over time. It designates, represents, or evokes a character, a culture, or an event from the past. The object is full of meanings, the same ones that change over time. By acquiring new definitions, the entity acquires a new value. It bases the cosmovision of the peoples by witnessing their history. This is a value deeply rooted in the vernacular housing of "the three spaces," and that is intended to be coined in heritage housing. (Validated by experts between 80% to 90% of high and medium incidence in the proposal).

## 4. Discussion

### 4.1. The Term Vernacular Housing

As seen in annex 2, there is a diversity of terms and concepts associated with vernacular housing. Multiple definitions of vernacular architecture have been generated, such as autochthonous architecture (that was born or originated in the same place where it is

located), famous (belonging to or related to the town), traditional (following the ideas, norms, or customs of the past), as well as the vernacular (domestic, native, of our house or country), which encompasses the definitions previously described [28]. One can also speak of cultured and popular architecture [46]. Other authors establish a classification model based on the essential characteristic components of architectural objects and according to elements such as style, form, ornament, structure, function, and the differential space of vernacular, rural, and urban architecture between others [29,47].

Homogeneity usually characterizes vernacular housing in forms, by the type of materials used, the construction system used, and obviously by the construction tradition. Communities transmitted the knowledge associated with these constructions from generation to generation throughout the history of these societies [48]. This oral tradition of transmission of empirical knowledge allows this type of architecture to be considered its "own" in each region in which it develops—achieving its identity and belonging to the area where the construction is located. There are examples of communities with vernacular housing with international recognition for their cultural values [49,50].

We found a wide diversity of concepts on vernacular housing in the literature. However, their analytical capacity did not allow addressing the problem raised in this investigation. Meanwhile, the ideas reflect the features and characteristics of vernacular housing but do not offer space for technical innovation. That is why the authors transitioned towards a concept of heritage housing based on bioclimatic criteria and sustainability, that is of prioritized social interest for the study region.

Under this approach, in the present investigation, a new Biosuvernacular Manabita housing design was provided that is based on the following principles: (1) respect for nature and the local environment; (2) better quality of life and the health of its inhabitants; (3) use of indigenous and renewable construction materials; (4) it is inserted in the landscape; (5) it uses labor and ancestral techniques in its construction; and (6) it mitigates risk factors. We made a comparison with a graph of the characteristics of the housing of the three spaces of 1736, the current housing, and the Biosuvernacular housing (see Table 3).

Moving analytically to the Biosuvernacular housing concept makes sustainable solutions to the housing problem in rural environments, aligning and making it more viable to achieve the indicators and goals of the 2030 agenda and sustainable development objectives.

*4.2. The Proposal for a New Design of the Houses of the Three Spaces with Sustainability Criteria and Their Conservation to Be Declared as Cultural Heritage*

There are different proposals for sustainable housing in the scientific literature and professional practices. In countries such as Spain and El Salvador, projects that propose to design an urban housing model following bioclimatic design and sustainability criteria have been published [51,52].

In Colombia, the VISS Sustainable Social Interest Housing and the VIPS Priority Interest Housing are located as housing intended for people with lower income [31]. Similar proposals for the urban setting exist [53]. There are also proposals to use the intelligent solar house that seeks to improve the quality of life without increasing energy consumption [54,55].

Bamboo and similar cane are recommended as a sustainable way to build affordable houses that withstand earthquakes. This renewable resource is often combined with other materials, such as straw, ceramic tiles, and concrete to build resilient structures [56]. Another typology uses concrete in the columns and clay roof tiles, along with the "Timagua" option [57]. Venezuelan designers note the central role of bamboo for sustainable buildings [58,59].

Other experiences use wood as a sustainable material in the construction of houses. [60] A prototype is proposed for the Mojana region in Colombia to mitigate the effects of floods. The drawback of this experience is that the wood takes a long time to reach its mature age. For example, teak takes 25 years to reach its cutting maturity, while bamboo regrows rapidly [61].

**Table 3.** Variables and indicators of the Biosuvernacular dwelling in relation to the home of the three spaces. Source: by the author.

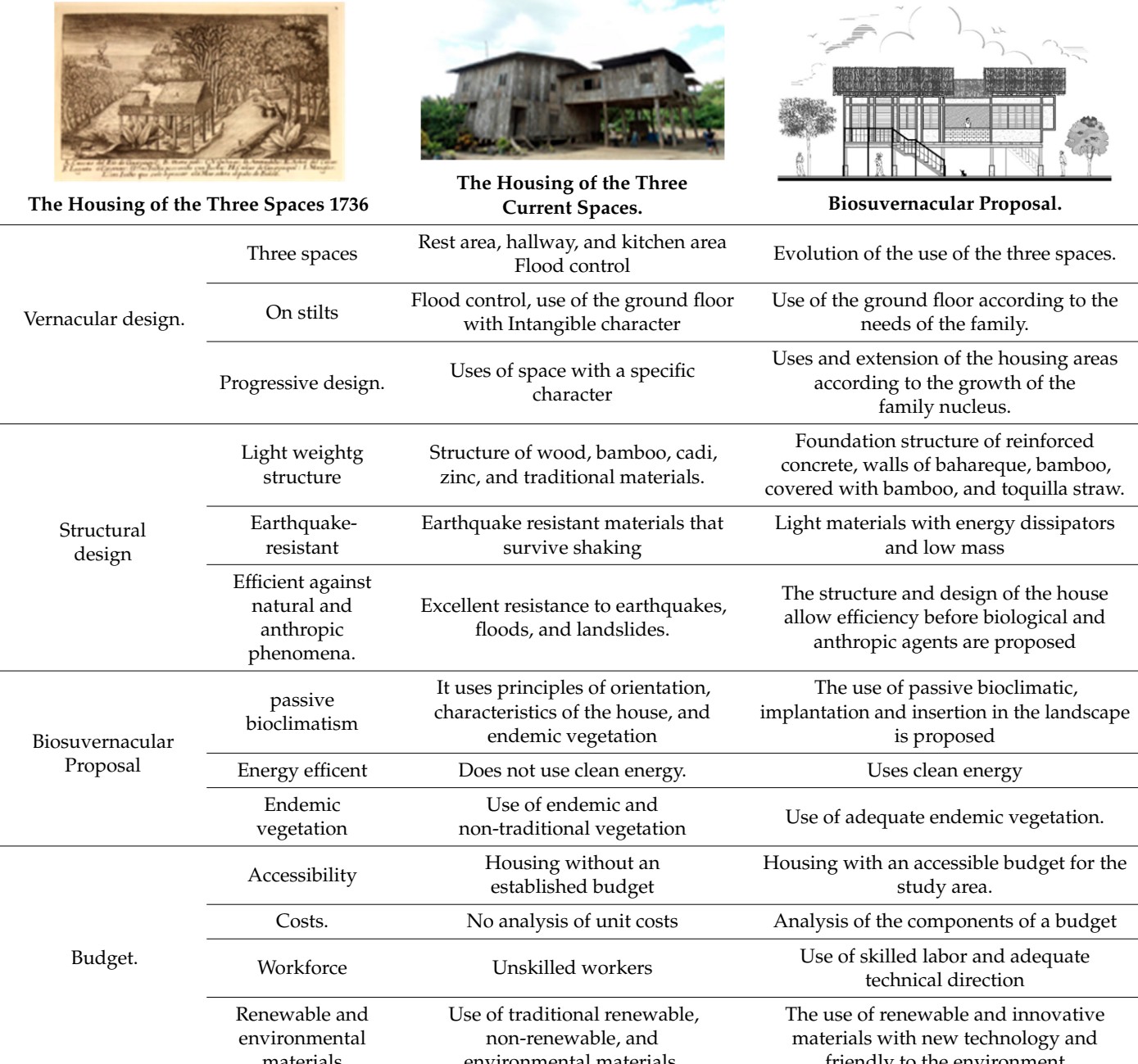

| | | The Housing of the Three Spaces 1736 | The Housing of the Three Current Spaces. | Biosuvernacular Proposal. |
|---|---|---|---|---|
| Vernacular design. | Three spaces | | Rest area, hallway, and kitchen area Flood control | Evolution of the use of the three spaces. |
| | On stilts | | Flood control, use of the ground floor with Intangible character | Use of the ground floor according to the needs of the family. |
| | Progressive design. | | Uses of space with a specific character | Uses and extension of the housing areas according to the growth of the family nucleus. |
| Structural design | Light weightg structure | | Structure of wood, bamboo, cadi, zinc, and traditional materials. | Foundation structure of reinforced concrete, walls of bahareque, bamboo, covered with bamboo, and toquilla straw. |
| | Earthquake-resistant | | Earthquake resistant materials that survive shaking | Light materials with energy dissipators and low mass |
| | Efficient against natural and anthropic phenomena. | | Excellent resistance to earthquakes, floods, and landslides. | The structure and design of the house allow efficiency before biological and anthropic agents are proposed |
| Biosuvernacular Proposal | passive bioclimatism | | It uses principles of orientation, characteristics of the house, and endemic vegetation | The use of passive bioclimatic, implantation and insertion in the landscape is proposed |
| | Energy effecent | | Does not use clean energy. | Uses clean energy |
| | Endemic vegetation | | Use of endemic and non-traditional vegetation | Use of adequate endemic vegetation. |
| Budget. | Accessibility | | Housing without an established budget | Housing with an accessible budget for the study area. |
| | Costs. | | No analysis of unit costs | Analysis of the components of a budget |
| | Workforce | | Unskilled workers | Use of skilled labor and adequate technical direction |
| | Renewable and environmental materials | | Use of traditional renewable, non-renewable, and environmental materials. | The use of renewable and innovative materials with new technology and friendly to the environment |

In the state of Potosí, Mexico, the objective is to demonstrate the degree of sustainability of rural housing with a construction tradition using stone and reed in walls, sheet roofing, and reed fences [62]. A separate point is the construction proposal's approach in rural El Salvador communities where they suggest using raw materials such as adobe for the construction of houses, using zinc as a cover material, reducing the heat loading to the house [63].

In Ecuador, vouchers are proposed to support public and private urban housing development. The last governments in power, on a smaller scale, proposed vernacular housing solutions of an emerging or rural nature and various housing constructions of all kinds that seek to solve the housing deficit in rural areas that currently exceeds 39% of a population of 5,600,000 as a result of floods and earthquakes. The national government,

through the Ministry of Housing (MIDUVI), has assigned bonds for the construction of new homes of USD10,000 and bonds for home improvement of up to USD4000.

Portoviejo, before and as a result of the earthquake of 16 April 2016, has implemented some housing alternatives to solve the problem of the growth of the housing deficit in the rural and urban parish sectors (see Table 4). INBAR [64] promotes this type of housing with reinforced concrete panels and a zinc roof, built in modular fashion, with three bedrooms, a bathroom, kitchen, and tanks for drinking water and wastewater. The house has a surface area of 64 m$^2$.

The Provincial Council of Manabí (CPM) [65] built a bahareque housing prototype using native materials from the Portoviejo area, a project developed by the architect Jorge Morán Ubidia. The Eloy Alfaro de Manta University (ULEAM) [66] proposed a housing prototype using guadúa cane for homes on the coast, trying to make it economically accessible. The Vice Presidency of the Republic and the technical secretary for Inclusive Management in Disabilities [67] proposed a house with accessibility. The corporation "Hogar de Cristo" [68] has been dedicated for 45 years to making houses in bamboo cane to solve the housing deficit in rural and marginal areas of large cities on the Ecuadorian coast.

Low-cost housing designs are also planned in Cuenca [69]. In the absence of Ecuadorian technical codes that meet the principles of ecological design, these use the principles of Abdel and Aboulgheit [70]. These design strategies consider the fundamentals of an eco-home and the maximum use of solar radiation with a social character. They are designed on two floors with the possibility of expanding the bedrooms. Building materials combine wood, concrete, or bricks—these last two are energy consumers. In addition, the use of photovoltaic panels and rainwater accumulators is proposed. However, this design is not appropriate for rural areas due to the use of conventional materials that are energy consumers, and its composition is typical of the Ecuadorian highlands.

Likewise, another author proposes an affordable home for the Sierra region seeking to establish the relevance of the materials used and the most appropriate architectural solutions for the rural environment. However, this prototype uses heat-capturing materials such as reinforced concrete, fiber cement, and steel. They are all high-energy consumers. This type of housing is more recommended for places with low temperatures [71–73].

There are also emerging solutions with little construction time carried out by foundations or non-governmental organizations, such as the Hogar de Cristo Foundation [68]. These propose housing solutions with little expectation of duration. They use materials such as bamboo, wood, concrete, and zinc on the roof. They meet minimum habitability criteria with tiny spaces but are destined to fail in a decade or so.

All the proposals pursue the objective of presenting designs that apply sustainability in the creation of housing that can be built in a short time. However, 25% of those analyzed are designed using non-renewable materials, and 75% include using renewable materials that allow a sustainable design.

All the designs proposed the implementation of different criteria such as climate, environment, physical well-being of the inhabitants, economic sustainability, optimum health, and structural safety. However, only one raises flood control as a variable to be considered, and none of them address the issue of culture and belonging at any time during the development of the design.

In Ecuador, the revised designs aim to solve short-term problems. After the 2016 earthquake, the demand for housing grew. The solutions proposed to solve this problem offer the use of conventional construction materials. The variables of heritage conservation and belonging and cultural identity are not considered. Nor is the use of renewable materials considered, as it is thought that using such denotes "low social status".

The solution proposed in this Biosuvernacular housing investigation incorporates Ecuadorian construction standards. This combines new technologies with ancient techniques in materials and craft. The design proves economical due to its "endemic" use of materials. Its light structure and elevated construction mitigate threats from floods and landslides.

**Table 4.** Housing alternatives proposed in Portoviejo after the 2016 earthquake. (Source: prepared by the authors). Define here DISENSA, etc.

| Entity | Type of Housing | Material Used | Image |
|---|---|---|---|
| DISENSA | 1 floor. 64 m$^2$ | Reinforced concrete |  |
| CPM | 1 floor | Bahareque and Bamboo |  |
| ULEAM | 1 floor | Bamboo |  |
| MIDUVI. | 1 floor. 42 m$^2$ | Reinforced concrete |  |
| GOBIERNO ECUADOR | 1 folor. 47, 50 m$^2$ | Reinforced concrete |  |
| HOGAR DE CRISTO | 2 floor. 36 y 42 m$^2$ | Bamboo and wood. |  |

Our study revealed the relevance of updating the vernacular housing designs of the three spaces while preserving the original cultural and environmental values. This update established the bases for the future preparation of a file to request its declaration as the cultural heritage of Ecuador.

## 5. Conclusions

This research revealed the cultural value of the houses in the three spaces and their potential to implement new strategies related to developing sustainable habitats in rural communities, which can be extended to similar realities in Latin America.

The article promoted the rescue of cultural values related to the rural habitats of Portoviejo, which have treasured their populations and have been reproduced from generation to generation. This research promotes continuity from tradition to modern and sustainable technical proposals before the region's environmental challenges.

A new analytical concept of Biosuvernacular housing was provided, which marks the transition from Manabita vernacular housing to a more sustainable and modern one, preserving its values and traditions.

The Biosuvernacular house integrates heritage values with modern and economical solutions, offering a budget to the demanding population and built by trained technical personnel. This design responds to the housing needs of rural parishes in Ecuador. These guidelines raise the need to include the results of scientific research in heritage management, which opens an essential line of socio-anthropological and architectural research in the region.

The adaptive capacity of the new Biosuvernacular housing proposal to face the socio-economic and environmental risks of vulnerable populations in the region was revealed. This research opens the opportunity to develop new lines of research focused on the following questions: How does one rethink the vernacular tradition in the face of economic and environmental challenges that allow proposing viable solutions in the current rural context? And how is it possible to guarantee the right to a sustainable habitat aligned to the SDGs for low-income populations who bear traditional cultural values?

This research provides a systematic approach to understanding the physical, cultural, and traditional elements required to create new sustainable housing for communities in Ecuador. The methodology used in this research can be replicated for other analyses of vernacular homes in the world.

**Author Contributions:** Conceptualization, R.V.H.Z., C.B.M. and O.P.M.; methodology, R.V.H.Z., C.B.M. and O.P.M.; software, C.M.-R.; validation R.V.H.Z., C.B.M. and C.M.-R.; formal analysis, R.V.H.Z. and C.B.M.; investigation, R.V.H.Z., C.B.M. and O.P.M.; resources, R.V.H.Z. and C.B.M.; L.O.N.B., D.C.L. and R.G.G.F.D.V.; data curation, C.M.-R. and R.V.H.Z.; writing—original draft preparation, R.V.H.Z., C.B.M. and O.P.M.; writing—review and editing, C.B.M. and B.C.; visualization, C.B.M.; supervision, C.B.M. and O.P.M.; project administration, R.V.H.Z., C.B.M. and O.P.M.; funding acquisition, C.B.M. and R.V.H.Z. All authors have read and agreed to the published version of the manuscript.

**Funding:** This research was funded partially by the Universidad de la Costa by the project entitled "Methodology for the use of renewable native materials in rural and coastal areas of Ecuador for construction purposes" (index project, INV code 1106-01-006-13) and by the Cuban project (PN 211LH012-018) "Adaptive Governance for the coastal and marine planning in Cuba", Universidad de Oriente, Cuba.

**Institutional Review Board Statement:** The study was conducted in accordance with the Declaration of Helsinki, and approved by the Institutional Review Board (or Ethics Committee) of Universidad Técnica de Manabí (Official Letter No: UTM II 2018-011-OF, 25 January 2018) for studies involving humans.

**Data Availability Statement:** No applicable.

**Acknowledgments:** The authors thank the Universidad de la Costa by the project entitled "Methodology for the use of renewable native materials in rural and coastal areas of Ecuador for construction

purposes". (Index project. INV code 1106-01-006-13). Thank you to the Universidad Técnica in Manabí, Universidad de la Costa, in Barranquilla, and Universidad de Oriente for the support.

**Conflicts of Interest:** The authors declare no conflict of interest. The funders had no role in the study's design, in the collection, analyses, or interpretation of data; in the writing of the manuscript; or in the decision to publish the results.

## Appendix A

**Table A1.** Conceptual revision regarding the vernacular term.

| Term | Author | Concept | Relevant Aspect |
|---|---|---|---|
| Vernacular architecture | Bernard Rudoksky, 1964 (cited by: Tillería 2011) [28] | What started as a series of examples of exotic buildings has been transformed into a recognized heritage category. | Vernacular architecture is carried out in the community by ordinary people rather than experts. |
| | Dictionaire historique de la langue Robert, (1985) [74] | It is derived from the term verna house, which in Latin means "slave born in the home," and vernacular means "indigenous" or "family"; the concept was coined by Roman law by Emperor Theodosius the Great (347–395) in the fourth century S.D.G. | Arguably, they were the first conceptualizations of "vernacular". This concept stems from the nature of society, the house, and home. |
| | Vela Cossío (1995) [75], | He sees the vernacular model as the result of generational collaboration and empathy between the craftsman who built it and the demanding user. | Knowledge is passed orally from generation to generation with the consent of the users. |
| | "Charter on the built vernacular heritage (1999)" by ICOMOS (1999) [3]. | Vernacular architecture is "a fundamental expression of community identity, in its relationship to the territory, and at the same time an expression of the world's cultural diversity". | The main features of cultural architecture are architecture by professionals and famous and popular architecture made by empiricists through experience gained through oral traditions. |
| | (Rapoport 2011) [1] | It is an architecture without theoretical or aesthetic conceit. It is related to a place, a people, and a tradition. It defines the identity of a territory and the elements of its cultural differences. Building without an architect. | It is defined as informal buildings that become part of the community's identity based on the builder's or owner's taste. |
| | Huelva y Ans, (2013) [34] | Vernacular architecture has unknowingly been implicitly passive strategies to respond to the environment. This is the case with constructive solutions using straw on the roof, where different cultures of the world (American, Asian, African, and European) have developed similar constructive solutions without cultural contact. (p. 104) | Vernacular architecture is a response to protecting man from natural elements such as weather and building using elements from his environment. |
| Urban vernacular architecture | Maldonado (2003) [47] | "Meeting the basic need for conservation, made from existing materials, whether industrialized or obsolete", p. 21 | The use of non-renewable materials that affect climate change is recommended. |
| Contemporary rural housing | Lizancos, (2005) [76] | It includes various periods from 1959 to the present: ancient, ruptured, and involute. | It showcases rural architecture as a sustainable model based on building materials. |

**Table A1.** *Cont.*

| Term | Author | Concept | Relevant Aspect |
|---|---|---|---|
| Social interest housing sustainable | Bedoya, (2008) [31] | It is technically and economically feasible due to its environmental viability. Different techniques and materials are experimented with and added to the architectural design of high environmental quality. | Guaranteed houses that meet the following conditions: low cost, energy efficiency, high environmental quality, the use of ecological materials, and the use of urban services are considered sustainable and socially beneficial. |
| Urban vernacular housing | Maldonado (2009) [29] | Vernacular architecture has always existed. In the first part of the text, Maldonado identifies two types of vernacular architecture: rural and urban. The classification depends on where the object was built and what materials it was built with. | The urban and rural vernacular housing have the same characteristics, they use materials from the environment, they are made by self-construction, therefore didactic, empirical and influential, |
| Social interest housing | Decreto 2190 (2009) Artículo 2 República de Colombia [30] | One that combines elements that ensure its habitability, urban, architectural, and building design quality standards, up to a maximum of 135 months of the statutory minimum wage | Government-funded housing programs. At least 200 units each to enable large-scale construction and meet housing needs. |
| Traditional house | Huelva y Ans, (2013) [34] | This is designed by using passive measures. They are influenced by geographic location and climatic environment, faced with different solutions for traditional buildings, after centuries of trial and error, obtained effective and appropriate building design strategies for each climate type. | Traditional housing is used as protection against natural and human elements, manufactured by unskilled labor over time, and designed based on experience. |
| Contemporary rural housing | Huelva y Ans, (2013) [34] | A new type of single-family home. It originated in Galicia in northwestern Spain in the mid-20th century. It envisages a housing model without historical origins, different living needs, building techniques, and industrialized building materials. (Transition from "Ancient" Society to "Contemporary" Society, p. 107.) Comfort requirements and the use of new materials and construction resources have changed. Methods for existing computer applications such as LIDER1.0 can be adapted. (Energy Demand Constraints Department of Housing, 2006); DOE-2 Energy Modeling Program (U.S. Department of Energy, 2009), etc. | It allows the use of industrialized materials such as glass, brick, etc.; energy consumers and causes of $CO_2$ emissions. |

**Table A1.** *Cont.*

| Term | Author | Concept | Relevant Aspect |
|---|---|---|---|
| Social housing | Martínez et al. (2016) [35] | The one that consists of basic spaces such as 2 bedrooms, living room, dining room, and kitchen, as well as a full bathroom | A space consisting of basic spaces such as two bedrooms, living room, dining room, and kitchen, and a full bathroom |
| | Pávez y Mesa (2017) [36] | They are designed according to the creation of a priority program that promotes sustainable buildings. Sometimes it seeks to update homes to provide greater comfort to residents thermally. | The use of masonry and concrete as energy-consuming elements during preparation, use, and subsequent maintenance is recommended to improve the home's thermal comfort. |
| Subsidized housing | Hidalgo Dattwyler, Rodrigo; Alvarado Peterson, Voltaire; Jiménez Barrado, Víctor (2018) [77] | High presence of the state in its design and construction. Ancillary structures provide broad leeway for private actors to engage as fundamental pillars of housing supply. | The peri-urban areas of the city are fragmented, migrating to small communities, which allows the creation of sustainable infrastructure projects from a natural and economic point of view. |
| Vernacular construction | Herrera et al. (2019) [38] | Use high-efficiency materials such as cane, wood, straw fibers, and mud. | Renewable materials are recommended, with minimal ecological footprint and pollution during construction. |
| Sustainable modular housing | Herrera et al. (2019) [38] | It uses highly efficient local materials such as cane, wood, and straw fibers. In addition, it uses a material consisting of clay, silt, fine sand, and coarse sand (mud) to coat quincha panels for this type of house. | These materials ensure adequate structural performance, stability, and shock resistance. |
| Social interest sustainable housing. | Vanegas Ospino (2019) [39] | Proposal of guidelines for the architectural design and construction of a VIS prototype for Medellin, which includes sustainability criteria in its design based on the base line prepared and applicable to Medellin and the Metropolitan Area | Advise on architectural design and construction guidelines for the VIS prototype in Medellin, including sustainability criteria in the design based on baselines prepared and applicable to Medellin and the metropolitan area |
| Sustainable housings | Álvarez, y Zulueta, (2021) [78] | The following factors affect its development: 1. Bioclimatic design of the house, 2. Marketing actions (p. 374), 3. Environmental awareness, 4. Popular Economy, 5. Plaintiff contact, 6. Related Information 7. Sustainable housing supply 8. Image of the sustainable housing program (p. 375) | Proposes sustainable housing as a savings mechanism in the household economy that improves the environment; as available resources or building materials are used to reduce energy consumption in a comfortable home, it supports environmental sustainability. |

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
