# Peer review of "A Sustainable Proposal for a Cultural Heritage Declaration in Ecuador: Vernacular Housing of Portoviejo"

_sustainability, doi:10.3390/su15021115_

Round 1

Reviewer 1 Report

The paper presents a proposal for the sustainable retrofit of vernacular buildings. Define what a proposal for a cultural heritage declaration means. Please, use an image for describing the methodology. In the present way is only less comprehensible. Also, characteristics and similarities among vernacular dwellings mus be defined better referring to the literature. For example, the cluster analysis is used for defining the characteristics and similarities among hundreds of vernacular dwellings (refer to https://doi.org/10.26868/25222708.2019.210346 to considering also this aspect). This could help also for specifying better the type selection despite the territorial differences between Italy and Ecuador. The concept of the guideline you proposed is not clear. Which kind of guideline are? Have Jenn really published? Have Ben. Discussed with stakeholders? Describe better this part to suggest the novelty of your approach. Describe also in which way your results are replicable in other countries. 

Author Response

I am sharing our manuscript entitled “A sustainable proposal for a cultural heritage declaration in Ecuador: Vernacular housing of Portoviejo.” We thank your revision and helpful comments, which have helped improve our manuscript.

You can find in the attachment our responses.
Best regard, 

Reviewer 2 Report

The authors obviously spent a lot of time to compile potentially relevant data from a variety of sources. This seems already a substantial achievement. Eventually, model and variable selection procedures can always be challenged. Overall, I consider the selected factors to be promising. However, there are two aspects from my perspective:

I would like propose to change the title  

“A sustainable proposal for a cultural heritage declaration in Ecuador - Vernacular housing of Portoviejo”.

Author Response

Dear reviewer

I am sharing our manuscript entitled “A sustainable proposal for a cultural heritage declaration in Ecuador: Vernacular housing of Portoviejo.” We thank your revision and helpful comments, which have helped improve our manuscript. You can find in the attachment our responses.
Best regard, 

Reviewer 3 Report

Thank you for the possibility to review the paper and thereby get a glimpse regarding to the vernacular housing in Ecuador. The article is interesting, the researched problem has scientific potential, and may be of interest to potential readers. Although I evaluate the article positively, I suggest a few more additions:

1.      I would recommend you to broaden the literature review (introduction).

2.      Specific objectives and research hypotheses based on the literature review must be presented.

3.      Conclusions must be more in-depth and objective.

4.      Indication of future research should provide more clues for future research.

5.      Large tables (eg. 3) are chaotic. Try rearranging them.

The article presents scientific value and can be published after carefully reviewing the reported issues.

Author Response

(The authors gave the same response as above.)

Reviewer 4 Report

Dear authors,

The topic of the article is interesting and of current interest. The approach to the topic is appropriate and rigorous.

I just have a few suggestions related to the design of the article.

1. Figure 1, c): the map is not clear, find a better resolution image

2. It is not clear what is the connection between Figure 2 and the sentence "Four hundred thousand eight hundred seventy-nine housing units were registered in Manabí" to which it seems to refer. Specify what Figure 2 represents in this context

3. Line 158: (Figure 2), not (Figue 2)

4. In Table 1, translate the table header into English. Maybe it could also use more vertical image spacing

5. Lines 415-421: Check the numbering of the guidelines, in the image there are four, in the text from the second go directly to the fourth

6. Clarify the Table 3 with some intermediate horizontal lines, as it looks now it is very difficult to go through

7. Check the writing of References, the instructions in the journal templates are not followed everywhere

I think the article merits to be published, success!

Author Response

Dear reviewer
First, we want to thank you for your valuable time invested in checking our manuscript and for the helpful comments. we are sending in the attachment our responses.
Best regard

Round 2

Reviewer 1 Report

-